# NeuralPlane: An Efficiently Parallelizable Platform for Fixed-wing Aircraft Control with Reinforcement Learning

**Chuanyi Xue**\*, **Qihan Liu**\*, **Xiaoteng Ma**\*, **Yang Qi**, **Xinyao Qin**,

**Yuhua Jiang**, **Ning Gui**, **Jinsheng Ren**, **Bin Liang**, **Jun Yang**†
Department of Automation, Tsinghua University
{xcy22, lqh20}@mails.tsinghua.edu.cn
pony.xtma@gmail.com
yangjun603@tsinghua.edu.cn

## Abstract

Reinforcement learning (RL) demonstrates superior potential over traditional flight control methods for fixed-wing aircraft, particularly under extreme operational conditions. However, the high demand for training samples and the lack of efficient computation in existing simulators hinder its further application. In this paper, we introduce NeuralPlane, the first benchmark platform for large-scale parallel simulations of fixed-wing aircraft. NeuralPlane significantly boosts high-fidelity simulation via GPU-accelerated Flight Dynamics Model (FDM) computation, achieving a single-step simulation time of just 0.2 seconds at a parallel scale of $10^6$ aircraft, far exceeding current platforms. We also provide clear code templates, comprehensive evaluation and visualization tools, and hierarchical frameworks for integrating RL and traditional control methods. We believe that NeuralPlane can accelerate the development of RL-based fixed-wing flight control and serve as a new challenging benchmark for the RL community. Our NeuralPlane is open-source and accessible at `https://github.com/xuecy22/NeuralPlane`.

## 1 Introduction

Unmanned Aerial Vehicles (UAVs) equipped with autonomous flight control systems have found extensive application in various tasks such as crop protection [1], pipeline inspection [2], topographic mapping [3] and environmental monitoring [4], particularly in dangerous or inaccessible environments. Fixed-wing aircraft, in contrast to the commonly used multi-rotor drones, have demonstrated broad utility in these missions owing to their long flight endurance and high cruising speeds [5, 6, 7]. Nonetheless, the inherent complex nonlinearity in dynamic models and strong sensitivity to atmospheric disturbances make traditional control approaches inadequate for ensuring agile and stable flight under extreme operational conditions [8, 9, 10]. As a result, addressing the challenges in fixed-wing aircraft control necessitates the utilization of advanced techniques that can deliver rapid responsiveness and robust stability.

Reinforcement learning (RL) has made significant progress in recent years, achieving superhuman performance in various domains, particularly in areas characterized by complex nonlinear dynamics such as robotic manipulation [11], autonomous driving [12] and quadrotor flight control [13]. RL operates on the trial-and-error principle, where an agent interacts with the environment, adapts

---

\*Equal contribution
†Corresponding author

38th Conference on Neural Information Processing Systems (NeurIPS 2024) Track on Datasets and Benchmarks.

continuously based on feedback, and optimizes decision-making to maximize cumulative rewards [14]. In the field of fixed-wing aircraft control, the potential benefits of applying RL are manifold. The complex nonlinear dynamics can be treated as a black-box model and addressed through exploration and exploitation. Moreover, the adaptive nature of RL enables systems to adapt their behavior to novel environmental conditions, such as variations in air density, wind direction, and intensity.

Despite an increasing number of researchers discovering the potential of RL and conducting notable work on fixed-wing aircraft control [15, 16, 17, 18], significant obstacles hinder its wider application in complex control scenarios. One significant challenge is the substantial demand of training samples that RL algorithms require, making it prohibitively expensive to gather sufficient interaction data from real flights. The existing simulation environments for fixed-wing aircraft either lack high fidelity to bridge the sim-to-real disparity [19], or fail to support large-scale efficient parallelization [20, 21, 22, 23], thus limiting the scalability of RL methods for more complex tasks.

*Can we build a platform that supports large-scale high-fidelity parallel simulations of fixed-wing aircraft and is convenient for training, testing, and evaluating RL algorithms?*

To answer this question, we propose a GPU-accelerated Flight Dynamics Model (FDM) for fixed-wing aircraft dynamics, which significantly improves simulation efficiency at large-scale parallel simulations, far surpassing currently common fixed-wing aircraft simulation platforms. Building on this, we introduce NeuralPlane, the first benchmark platform supporting large-scale parallel simulations for fixed-wing aircraft. This platform also features a clear system framework and interfaces to support the training, testing, and evaluation of RL algorithms. It integrates various task scenarios to validate the performance of RL in controlling fixed-wing aircraft across different problem contexts. Furthermore, the simulation's high fidelity facilitates the transfer of trained RL algorithms to real-world scenarios. Our contributions can be outlined as follows:

1. **Support large-scale parallel high-fidelity simulation of fixed-wing aircraft**. We propose a GPU-accelerated FDM that supports large-scale parallel high-fidelity simulations. Our method achieves a single-step simulation time of 0.2 seconds at a parallel scale of $10^6$ aircraft, surpassing current platforms. Figure 1 shows the comparison results.

2. **Provide multiple fixed-wing aircraft task scenarios and baseline algorithms**. NeuralPlane includes various basic task scenarios for fixed-wing aircraft and allows researchers to customize tasks through provided interfaces. It integrates both traditional control methods and RL methods, offering detailed analysis and evaluation of their performance.

3. **Provide clear code templates and frameworks**. We offer code templates for RL and traditional control methods, along with interfaces for interacting with the simulation environment, enabling researchers to easily train, test, and evaluate their algorithms.

4. **Support algorithm evaluation and visualization of control results**. We propose various metrics to evaluate algorithm performance, making analysis convenient for researchers. We also support different visualization tools for rendering flight trajectories, to intuitively assess control results.

## 2 Related Work

**Fixed-wing Aircraft Control**   Traditional approaches [24, 25, 26] in fixed-wing aircraft control heavily rely on elaborately designed gain scheduling techniques with linearized plant dynamics(e.g. Proportional-Integral-Derivative (PID) control [26]). However, these methods have been found to be less flexible to changes in model dynamics, such as task redistribution and wind disturbances [8, 9, 10]. On the other hand, RL has been progressively integrated into this domain, enhancing aircraft control systems with the ability to optimize decision-making, develop robust policies, and adapt to unstable and noisy dynamics through feedback from simulation environments. The adaptive learning facilitated by RL in fixed-wing aircraft control can cover a range of tasks, such as attitude control [15], autopilot management [16], control surface coordination [17], and even the autonomous response to unexpected events without immediate human intervention [18].

**RL methods in Aircraft Control Challenges**   Various RL algorithms have been leveraged for diverse flight control tasks. For instance, Proximal Policy Optimization (PPO) [27] has been employed in attitude control to manage the complex nonlinear dynamics inherent in aircraft systems [15, 28, 17].

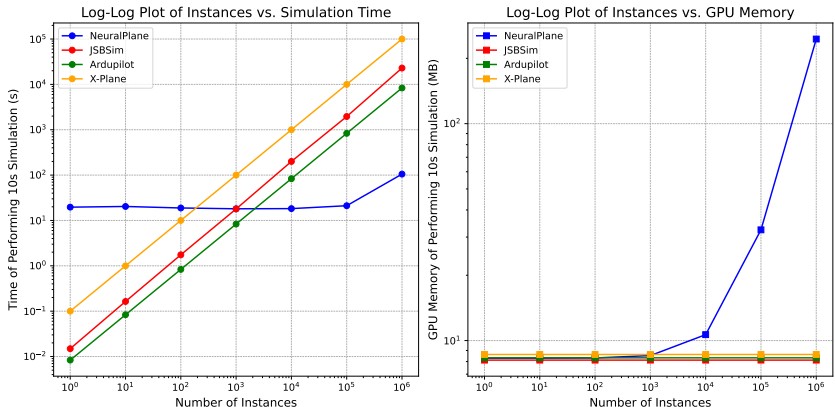

Figure 1: Performance comparison between NeuralPlane and several classic platforms. **Left**: Relationship between the time required to simulate 10 seconds and the number of parallel simulations. **Right**: Relationship between GPU memory usage for simulating 10 seconds and the number of parallel simulations. **Blue line**: Results for NeuralPlane. **Red line**: Results for JSBSim [21]. **Green line**: Results for Ardupilot [22]. **Orange line**: Results for XPlane [20].

Deep Deterministic Policy Gradient (DDPG) [29] has exhibited resilience to varying wind disturbance conditions in the context of automatic landing control for fixed-wing aircraft, especially when provided with well-designed reward functions [30]. Soft Actor-Critic (SAC) [31] has demonstrated superior performance over traditional controllers in both autonomous flight [32] and maneuver generation [18, 33], particularly in dealing with continuous action spaces. Beyond the field of online model-free RL, fixed-wing aircraft control challenges are increasingly recognized as critical benchmarks across various RL domains. Offline RL methods have been introduced to tackle challenges like off-field landings in unprepared locations [34] and enhance data efficiency in attitude control tasks [35]. Furthermore, tasks related to navigation [36] and path planning [37, 38] can be reconceptualized as Goal-Conditioned RL (GCRL) problems, thereby boosting the capacity for generalization. Additionally, the emerging challenges of traffic control [39, 40], collision avoidance [41, 42], and cooperative decision-making [43] within multi-aircraft systems are establishing themselves as stringent benchmarks for Multi-Agent Reinforcement Learning (MARL) [44, 45, 46, 47, 48, 49, 50, 51]. However, integrating RL into flight control has lots of challenges. Ensuring safety, achieving data efficiency, and sim-to-real transferring are central areas that require careful attention, particularly in complex tasks under extreme operational conditions [10].

**Aircraft Simulation Platforms** The development of aircraft simulation platforms has become a focal point within academia, primarily due to their essential role in the filed of flight control. XPlane [20], renowned for its accurate and realistic physics modeling, has been utilized in numerous research projects [52, 53]. However, due to its commercial closed-source nature, researchers must procure a real-time rendering game and install a UDP-based connector, XPlaneConnect [54]. The absence of a headless mode in XPlane, which would allow for simulations to run without graphical rendering, positions it as a more suitable tool for testing rather than training. ArduPilot [22], an open-source autopilot system, has been specifically designed for unmanned aerial vehicles (UAVs) and has garnered significant favor within drone communities due to its generality across flight modes. While originally tailored for autonomous flight, ArduPilot is mainly employed for remote control applications in quadcopters [55, 56, 57] and requires extra adaptations to directly interface with RL on fixed-wing control tasks. JSBSim [21] stands out as a highly flexible, open-source flight dynamics simulation platform that supports headless mode operation and integrates with the open-source FlightGear rendering software. While JSBSim presents itself as an entirely cost-free option for RL researchers [16, 17, 58, 59, 60], its reliance on CPU-based Flight Dynamics Model (FDM) computation, devoid of GPU acceleration, restricts its capacity for parallelized deployment. QPlane [23] proposes the first toolkit which combines RL training on fixed-wing flight with multiple flight simulators (XPlane and JSBSim). Despite inheriting the parallelization limitations from JSBSim, QPlane incorporates a standard Gym interface for reinforcement training. This integration facilitates flexible replacement of RL algorithms and has been effectively utilized in a variety of research

Table 1: A summary of related work on aircraft simulation platforms. NeuralPlane is the first work that incorporates designs from all four domains.

| Platform | FDM Computation | Headless Mode | RL Integration | GPU Acceleration |
|---|---|---|---|---|
| XPlane [20] | ✓ | ✗ | ✗ | ✗ |
| ArduPilot [22] | ✓ | ✓ | ✗ | ✗ |
| JSBSim [21] | ✓ | ✓ | ✗ | ✗ |
| QPlane [23] | ✗ | ✓ | ✓ | ✗ |
| MaCA [19] | ✗ | ✓ | ✓ | ✗ |
| NeuralPlane | ✓ | ✓ | ✓ | ✓ |

projects [61, 62]. MaCA [19] is another aircraft platform, which also integrates RL algorithms with heterogeneous multi-agent cooperative decision-making tasks, but treats flights as mass points without actual FDM computation. However, existing aircraft simulation platforms have only implemented standard RL interfaces based on Gym, lacking specialized acceleration designs for large-scale parallel simulation, a critical aspect highlighted in robotics manipulation by IsaacGym [63]. To the best of our knowledge, we are the first to develop the fixed-wing aircraft control platform supporting efficient large-scale parallelization with GPU acceleration for RL.

**Hardware-accelerated Simulation Platforms**    Running physics simulations on GPUs can lead to significant speedups, which is crucial for RL training. In recent years, hardware-accelerated simulation platforms have greatly advanced the RL field. For example, NVIDIA's Isaac Gym [64] enables high-performance training for various robotics tasks directly on GPUs. In autonomous driving, platforms like GPUDrive and Waymax offer similar benefits. GPUDrive [65], built on the Madrona Game Engine, can generate over a million steps per second, allowing for fast and effective RL training using the Waymo Motion dataset. Waymax [66], designed for large-scale simulation in multi-agent scenarios, further advances autonomous driving research.

Another example is Pgx [67], a suite of board game RL environments optimized for GPU/TPU accelerators, which simulates environments 10-100 times faster than existing Python implementations. These developments underscore the importance of building RL training platforms that support massively parallel simulations, as they can significantly enhance the speed and effectiveness of RL algorithms across various domains. We believe that NeuralPlane can also advance RL in the fixed-wing aircraft domain.

## 3   NeuralPlane: Design and Resources

### 3.1   Preliminaries

The control process of fixed-wing aircraft can be modeled as an MDP. An MDP can be defined by a tuple $\langle \mathcal{S}, \mathcal{A}, P, R, \gamma \rangle$. Define the state of fixed-wing aircraft at time step $t$ as $\mathbf{x_t}$, the control input as $\mathbf{u_t}$, and the task objective as $\mathbf{x_{target}}$. Then the observed state at time step $t$ can be expressed as $\mathbf{s_t} = normalize([\mathbf{x_t} - \mathbf{x_{target}}, \mathbf{x_t}])$, and the action at time step $t$ can be expressed as $\mathbf{a_t} = normalize(\mathbf{u_t})$, where $normailze$ denotes the normalizing function. The reward at time step $t$ can be expressed as $\mathbf{r_t} = d(\mathbf{x_t}, \mathbf{x_{target}})$, where $d$ denote the distance metric function. More details are given in the Appendix A.1.

### 3.2   Architecture and Workflow

The architecture and workflow of NeuralPlane are illustrated in Figure 2. The platform consists of three main modules: the *simulation environment*, the *baseline library*, and the *performance evaluator*. The *simulation environment* supports dynamic simulations of various fixed-wing aircraft and interacts with control algorithms. It includes interfaces for the aircraft model, environmental parameters, and the interaction between the environment and algorithms. The simulation environment supports large-scale parallel simulations, ensuring high computational efficiency and meeting RL training requirements for sample complexity. The *baseline library* includes traditional control methods and

RL algorithms. The *performance evaluator* proposes various metrics and different visualization tools to evaluate the performance of algorithms, making analysis convenient for researchers.

The workflow of the platform consists of three primary stages: *training*, *testing*, and *evaluation*. Before *training*, we first select the fixed-wing aircraft model, the task scenario, and the control algorithm. The algorithm is then trained on the platform in a fully automated process. After training, we conduct the *testing* stage under the specific testing scenario and compare with baseline algorithms. The platform automates the evaluation process and records flight data. Based on this data, the platform calculates performance metrics to evaluate the algorithm's control performance in the *evaluation* stage. Additionally, we can replay the flight data with different visualization tools to observe the algorithm's control results directly, facilitating further optimization.

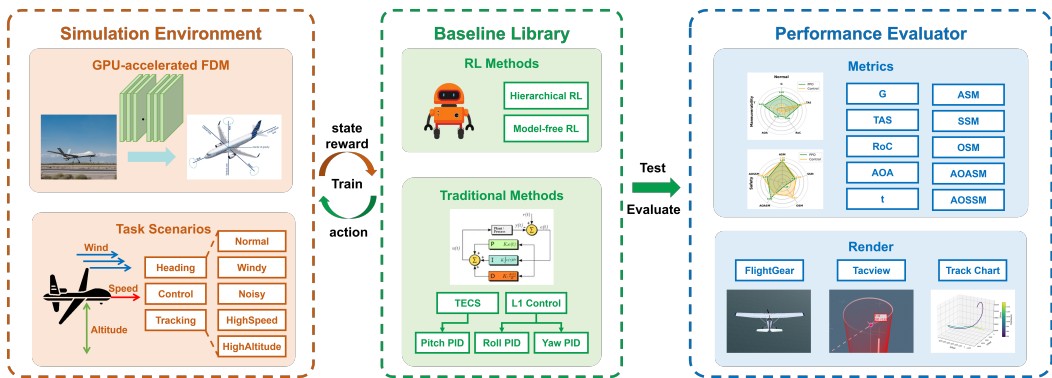

Figure 2: The overall architecture and workflow of NeuralPlane.

## 3.3 Simulation Environment

The simulation environment is the core component of NeuralPlane, distinguishing it from other fixed-wing aircraft simulation platforms with its capability for large-scale parallel simulations. This feature is crucial for meeting the sample complexity and data efficiency requirements of RL algorithm training. The simulation environment consists of two main modules: the GPU-accelerated FDM and the task Scenarios. These modules together provide a comprehensive and efficient training ground for advanced control algorithms. For more details, see the Appendix A.1.

**GPU-accelerated FDM** Large-scale parallel computations in the simulation environment are enabled by the parallel solution of fixed-wing aircraft dynamics equations. State variables, control variables, and aerodynamic parameters are stored in tensor format, allowing GPUs to accelerate tensor operations and improve efficiency. Traditionally, aerodynamic parameters are read from lookup tables, a process that hinders parallel efficiency due to extensive logical decisions. To address this, we use a multi-layer perceptron (MLP) to approximate these parameters, replacing table lookups with MLP predictions in practical model computations.

The simulation environment includes several classical fixed-wing aircraft dynamics models, such as the Cessna 172P and F16. It also features clear interfaces and documentation, enabling researchers to integrate their own fixed-wing aircraft models.

**Task Scenarios** Task scenarios in the simulation environment are defined by task objectives and flight conditions. Table 2 and Table 3 describe a brief introduction to several typical task scenarios integrated into the platform. For detailed content, see the Appendix A.2.

Table 2: Task scenarios categorized by objectives.

| Name | Target | Difficulty |
|---|---|---|
| Heading | altitude, yaw angle, and speed | easy |
| Control | pitch angle, yaw angle, and speed | middle |
| Tracking | coordinate position (geocentric coordinate) | difficult |

Table 3: Task scenarios categorized by flight conditions.

| Name | Key Value | Difficulty | Description |
|------|-----------|------------|-------------|
| HighSpeed | speed | middle | speed exceeding Mach 1 |
| Noisy | noise scale | middle | noisy observations |
| HighAltitude | altitude | difficult | altitude exceeding 30,000 feet |
| Windy | airspeed | difficult | airspeed not equal to 0 |

By combining different task objectives and flight conditions, NeuralPlane supports to set up various task scenarios. The platform also offers clear parameter setting interfaces, allowing researchers to configure different task objectives and conditions independently.

### 3.4 Baseline Library

NeuralPlane integrates two types of baseline algorithms: traditional methods and RL methods. Below is a brief introduction to these algorithms. For detailed content, see the Appendix A.3.

**Traditional Methods**   These are based on open-source fixed-wing aircraft control algorithms from the Ardupilot platform, using a hierarchical control approach. The upper layer includes the TECS controller [68], which manages the aircraft's total flight energy by adjusting throttle and pitch to maintain desired altitude and speed, and the L1 controller [69], which manages the flight path by adjusting roll and yaw to follow waypoints or desired path characteristics. The lower layer consists of an attitude loop controller using a dual-loop PID algorithm to control the aircraft's surfaces and achieve three-axis attitude tracking.

**RL Methods**   PPO [27] is a well-known RL algorithm that has been successfully applied in fixed-wing aircraft control [15, 28, 17]. We use PPO for Heading and Control tasks in fixed-wing aircraft, demonstrating high training efficiency. For the Tracking task, we use a hierarchical RL method: the upper-level algorithm converts the target location into desired pitch, yaw, and speed, while the lower level uses the trained PPO algorithm to control the aircraft's surfaces.

### 3.5 Performance Evaluator

NeuralPlane evaluates fixed-wing aircraft control performance using maneuverability and safety indicators. Maneuverability indicators assess control performance, while safety indicators evaluate control safety. Below are typical indicators, with all performance metrics normalized. For more details, see the Appendix A.4.

**Maneuverability Indicators**   1) G: Average G-force during flight. 2) TAS: Average True Air Speed during flight. 3) RoC: Average Rate of Climb during flight. 4) AOA: Average Angle of Attack during flight. 5) t: Average time to complete the task objective.

**Safety Indicators**   1) Altitude Safety Margin (ASM): The difference between the average flight altitude and the minimum safe flying altitude. 2) Speed Safety Margin (SSM): The smaller value between the absolute differences of the average flight speed from both the maximum and minimum safe flying speeds. 3) Overload Safety Margin (OSM): The absolute difference between the average G-force and the maximum safe G-force. 4) Angle of Attack Safety Margin (AOASM): The absolute difference between the average angle of attack and the critical safe angle of attack. 5) Sideslip Angle Safety Margin (AOSSM): The absolute difference between the average sideslip angle and the critical safe sideslip angle.

## 4   Benchmarking Study

NeuralPlane is a valuable platform for RL research, making training, testing, and evaluation of algorithms easy to implement. It facilitates the analysis of maneuverability, safety, and robustness in fixed-wing aircraft control. This section presents examples demonstrating NeuralPlane's application

in experimental research. We also test NeuralPlane's parallel performance, showing the importance of large-scale parallel simulations for controlling flight in fixed-wing aircraft.

## 4.1 Experimental Setup

In the following experiments, unless specified otherwise, we use the F16 fixed-wing aircraft dynamics model. The training parameters are set as follows: the maximum number of training steps (M) is $1.35 \times 10^9$, the number of parallel rollouts (n) is 3000, and the number of steps per rollout (m) in one iteration is 3000. All experiments are conducted on an NVIDIA A100 GPU with 80GB of memory. NeuralPlane is compatible with other platforms as well. For detailed settings of the simulation environment and algorithm parameters, see the Appendix B.1.

## 4.2 Platform Performance Analysis

One of the standout advantages of NeuralPlane compared to other fixed-wing aircraft simulation platforms is its support for large-scale parallel simulations. We test the parallel capabilities of NeuralPlane and compare it with several mainstream fixed-wing aircraft dynamics simulation platforms. Figure 1 shows that when the number of parallel simulations exceeds 1000, NeuralPlane's computation time is significantly superior to the other platforms, clearly demonstrating NeuralPlane's excellent performance in parallel simulations.

Table 4: MLP fitting results for some aerodynamic parameters.

| Name | test set $R^2$ | test set error |
|------|----------------|----------------|
| Cl | $0.9949 \pm 0.0009$ | $0.0051 \pm 0.0006$ |
| Cm | $0.9962 \pm 0.0007$ | $0.0042 \pm 0.0003$ |
| Cn | $0.9926 \pm 0.0011$ | $0.0067 \pm 0.0013$ |
| Cx | $0.9967 \pm 0.0004$ | $0.0030 \pm 0.0007$ |
| Cz | $0.9991 \pm 0.0001$ | $0.0014 \pm 0.0003$ |

To enable large-scale parallel simulations of fixed-wing aircraft, NeuralPlane uses MLP to fit aerodynamic parameter data tables, accelerating the lookup computation process. The fitting accuracy of the MLP directly affects the platform's simulation accuracy. We test the MLP's fitting performance comprehensively, with results shown in Table 4. To eliminate the impact of randomness on the MLP fitting results, we repeat the experiments with multiple random seeds and calculate the mean and standard deviation of the $R^2$ and error on the test set. The mean $R^2$ values for the MLP fitting of aerodynamic parameters are all above 0.99, with low standard deviations, demonstrating high fitting accuracy. This confirms the platform's high-fidelity simulation capabilities.

## 4.3 Training of RL Algorithms

We use two dynamics models, the F16 and an unmanned fixed-wing aircraft (UAV), to train the RL algorithms for all tasks and conditions in the baseline library. Full experimental results are provided in the Appendix B.2, with a subset (F16 model for the Heading task) shown in Figure 4. A comparison using a small UAV dynamics model for the Heading task is shown in Figure 3. These results demonstrate that NeuralPlane supports RL training across various aircraft models and tasks. Owing to large-scale parallel training, the RL algorithms converge quickly, with the average episode reward steadily increasing and showing minimal fluctuations. The PPO algorithm converges in about 100 iterations, taking around one day.

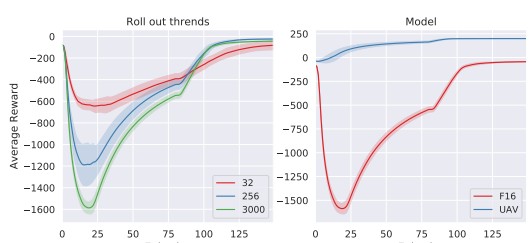

Figure 3: Training curves of PPO under different settings. **Left**: Experimental results with varying numbers of parallel rollouts during the training process. **Right**: Experimental results with different fixed-wing aircraft dynamics models.

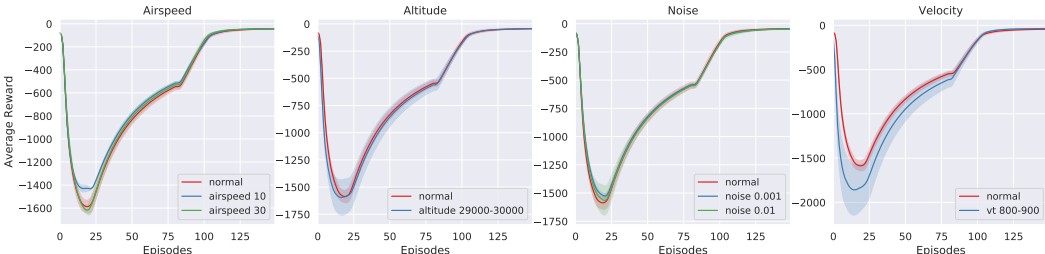

Figure 4: Training curves of PPO in different task scenarios. **From left to right**, the task conditions are different wind speeds, different flight altitudes, different environmental noise levels, and different flight speeds, with the task objective being the Heading task in all cases.

We also examine the effect of parallel rollout quantity on RL training, comparing results with n set to 32, 256, and 3000. The results, shown in Figure 3, indicate that with a small parallel training quantity, the RL algorithms do not converge and the average episode reward fluctuates significantly. This highlights the importance of NeuralPlane's support for large-scale parallel simulations.

## 4.4 Comparison of Different Baseline Algorithms

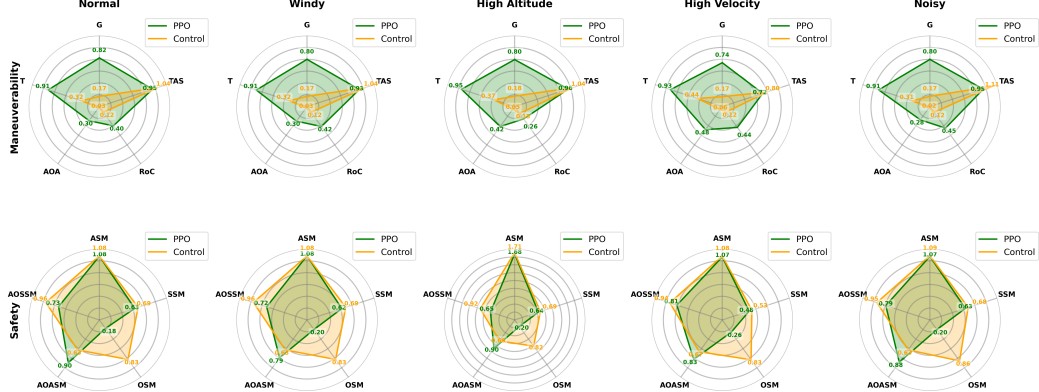

Figure 5: Comparison of control performance between traditional method and PPO in different task scenarios. **Green line**: PPO algorithm results. **Yellow line**: Traditional methods results.

Based on the Heading task, we compare the control performance of the traditional method and the PPO algorithm under different conditions (Normal, HighSpeed, HighAltitude, Windy, Noisy). Performance is evaluated using the proposed maneuverability and safety metrics. The results, shown in Figure 5, indicate that the PPO algorithm significantly outperforms the traditional method in maneuverability metrics under all conditions, especially in key indicators such as G, t, and RoC. In terms of safety metrics, the traditional method slightly outperforms the PPO algorithm. However, in any task scenario, the PPO algorithm's success rate in controlling the aircraft to reach the target is above 95 %, indicating that the safety of using the PPO algorithm to control fixed-wing aircraft is entirely acceptable. These experiments demonstrate the superiority of RL algorithms in controlling flights of fixed-wing aircraft.

## 4.5 Testing and Evaluation of Algorithms

NeuralPlane enables performance evaluation of algorithms for flights of fixed-wing aircraft using various metrics, and it also allows for the visualization and replay of flight data. This feature helps researchers intuitively assess algorithm performance. We design three visualization methods to aid in performance analysis: 1) using FlightGear for visual rendering of flight scenes; 2) using Tacview for visual analysis of flight trajectories and situations; and 3) plotting flight trajectories to directly

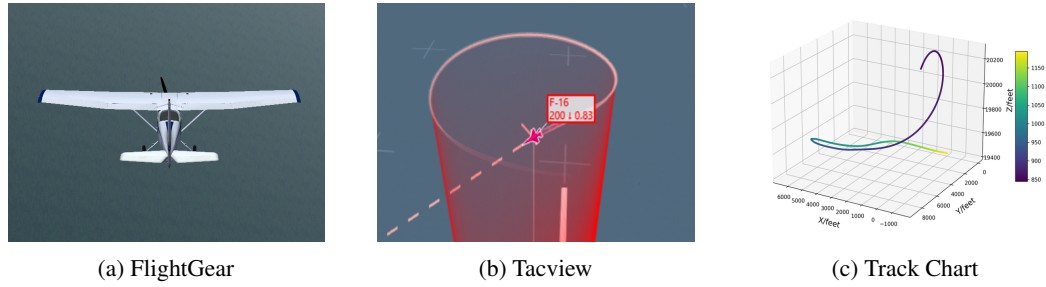

| (a) FlightGear | (b) Tacview | (c) Track Chart |

Figure 6: Visualization of fixed-wing aircraft flight trajectories.

analyze aircraft maneuverability. We demonstrate the visualization of trajectory of the PPO algorithm completing the Tracking task, as shown in Figure 6. Additional visualized flight trajectories are available in the Appendix B.2. This data replay mechanism in NeuralPlane aids researchers in algorithm design and debugging, enhancing the platform's usability.

## 4.6  Analysis of Algorithm Robustness

NeuralPlane includes methods for evaluating the robustness of algorithms. By testing algorithms under environmental noise disturbances and varying noise levels, researchers can compare control performance across different scenarios. This allows for analysis of the algorithms' robustness and disturbance rejection capabilities. The robustness metric is defined as the maximum noise level at which the algorithm can still control the aircraft safely, allowing for evaluation and comparison of different algorithms. The robustness analysis results, shown in Figure 7, indicate that when the noise scale increases to 0.01, the PPO algorithm can still complete the task, although the task completion time slightly increases. The results demonstrate that the PPO algorithm possesses a certain level of disturbance rejection capability.

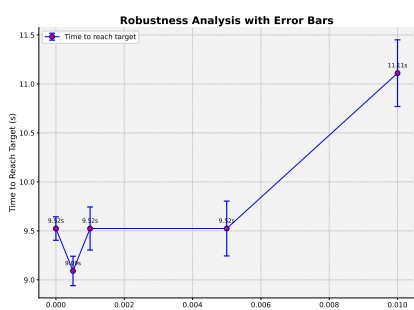

Figure 7: Robustness test results of the PPO algorithm.

## 5  Conclusion and Future Work

In this paper, we introduce NeuralPlane, the first benchmark platform for large-scale parallel simulations of fixed-wing aircraft, designed to advance the development of RL algorithms for flight control. Our platform addresses key challenges in existing simulation environments by offering GPU-accelerated FDM, achieving a single-step simulation time of just 0.2 seconds at a parallel scale of $10^6$, significantly outperforming current platforms.

Our experimental results demonstrate NeuralPlane's superior performance in large-scale parallel simulations, highlighting its efficiency and capability to train RL algorithms rapidly and effectively. Comparative analysis of PPO and traditional methods across various task scenarios reveals the superior maneuverability and acceptable safety performance of RL algorithms.

While our NeuralPlane presents a significant advancement, it also has several limitations that we aim to address. Currently, it supports a limited number of fixed-wing aircraft models and task scenarios. We plan to expand these to provide a richer set of tasks and functionalities for RL applications in fixed-wing aircraft control. Additionally, the platform does not yet support multi-aircraft scenarios. We intend to extend NeuralPlane to include these, facilitating multi-agent RL research. Finally, we aim to improve the platform's user-friendliness by designing clearer interfaces and workflows.

## Acknowledgments and Disclosure of Funding

This work was supported by the National Science and Technology Innovation 2030 - Major Project (Grant No. 2022ZD0208804).

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

# A    Details of Platform

## A.1    Flight Dynamics Model

The 6-DoF atmospheric dynamics of a rigid aircraft are described by a set of standard nonlinear ordinary differential equations, which are not detailed here for brevity; interested readers are referred to [9] [16]. This model differentiates between a ground-based inertial frame and an aircraft-based reference frame. The ground-based frame $\mathcal{F}_E = \{O_E; x_E, y_E, z_E\}$ is inertial, ignoring Earth's rotational effects, which is a valid assumption for low-altitude flight. The frame's origin is fixed at point $O_E$ on the ground, with $x_E$ pointing north, $y_E$ east, and $z_E$ downwards. This is also known as the NED (North-East-Down) frame. The aircraft body-fixed frame $\mathcal{F}_B = \{G; x_B, y_B, z_B\}$ originates at the aircraft's center of gravity $G$. Here, $x_B$ aligns with the fuselage pointing forward, $y_B$ points rightward, and $z_B$ downward.

The motion equations are derived from Newton's second law for an air vehicle, resulting in six core scalar equations (conservation of linear and angular momentum in $\mathcal{F}_B$), flight path equations (for tracking the aircraft's center-of-gravity relative to $\mathcal{F}_E$), and rigid-body kinematic equations (defining the aircraft's attitude quaternion to describe the body axes orientation relative to the inertial ground frame).

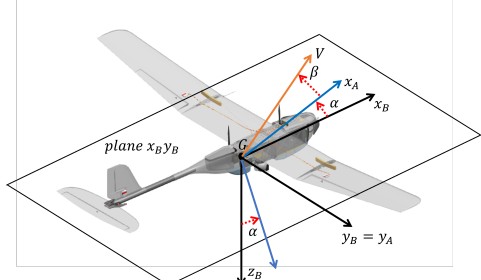

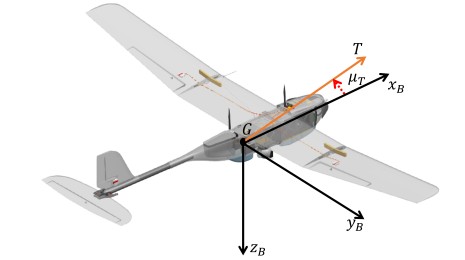

(a) Aerodynamic angles, aerodynamic (or stability) frame

(b) Thrust vector, thrust magnitude $T$, thrust line angle $\mu_T$

Figure 8: Fixed-Wing aircraft flight dynamics model

The conservation of linear momentum equations (CLMEs) for a rigid aircraft with constant mass can be expressed by the following three fundamental scalar equations 1:

$$\dot{u} = rv - qw + \frac{1}{m}\left(W_x + F_x^{(A)} + F_x^{(T)}\right) \tag{1a}$$

$$\dot{v} = -ru + pw + \frac{1}{m}\left(W_y + F_y^{(A)} + F_y^{(T)}\right) \tag{1b}$$

$$\dot{w} = qu - pv + \frac{1}{m}\left(W_z + F_z^{(A)} + F_z^{(T)}\right) \tag{1c}$$

where $\mathbf{W}$ represents the aircraft's weight, $F^{(A)}$ denotes the aerodynamic forces, and $F^{(T)}$ stands for the thrust forces. These forces are decomposed into body frame components $\mathcal{F}_B$ for simplicity in deriving Eqs. 1a, 1b, 1c.

The weight force, always aligned with the inertial $z_E$ axis, is $mg$ and its components in the body frame are given by:

$$\begin{Bmatrix} W_x \\ W_y \\ W_z \end{Bmatrix} = [T_{BE}] \begin{Bmatrix} 0 \\ 0 \\ mg \end{Bmatrix} = \begin{Bmatrix} 2(q_z q_x - q_0 q_y) \\ 2(q_y q_z + q_0 q_x) \\ q_0^2 - q_x^2 - q_y^2 + q_z^2 \end{Bmatrix} mg \tag{2}$$

The matrix $[T_{BE}]$ describes the direction cosines for the instantaneous attitude of frame $\mathcal{F}_B$ relative to frame $\mathcal{F}_E$. Its entries are functions of the aircraft's attitude quaternion components $(q_0, q_x, q_y, q_z)$ 3:

$$[T_{BE}] = \begin{bmatrix} q_0^2 + q_x^2 - q_y^2 - q_z^2 & 2(q_x q_y + q_0 q_z) & 2(q_x q_z - q_0 q_y) \\ 2(q_x q_y - q_0 q_z) & q_0^2 - q_x^2 + q_y^2 - q_z^2 & 2(q_y q_z + q_0 q_x) \\ 2(q_x q_z + q_0 q_y) & 2(q_y q_z - q_0 q_x) & q_0^2 - q_x^2 - q_y^2 + q_z^2 \end{bmatrix} \tag{3}$$

The aerodynamic force $F^{(A)}$ acting on the aircraft, projected onto frame $\mathcal{F}_B$, is given by 4:

$$\begin{Bmatrix} F_x^{(A)} \\ F_y^{(A)} \\ F_z^{(A)} \end{Bmatrix} = [T_{BW}] \begin{Bmatrix} -D \\ -C \\ -L \end{Bmatrix} \tag{4}$$

$$\begin{Bmatrix} F_x^{(A)} \\ F_y^{(A)} \\ F_z^{(A)} \end{Bmatrix} = \begin{bmatrix} -D \cos\alpha \cos\beta + L \sin\alpha + C \cos\alpha \sin\beta \\ -C \cos\beta - D \sin\beta \\ -D \sin\alpha \cos\beta - L \cos\alpha + C \sin\alpha \sin\beta \end{bmatrix} \tag{5}$$

The aerodynamic drag $D$, cross force $C$, and lift $L$ account for the effects of external airflow. The coordinate transformation matrix $[T_{BW}]$ from the standard wind frame $\mathcal{F}_W = \{G; x_W, y_W, z_W\}$ to $\mathcal{F}_B$ is given by:

$$[T_{BW}] = \begin{bmatrix} \cos\alpha & 0 & -\sin\alpha \\ 0 & 1 & 0 \\ \sin\alpha & 0 & \cos\alpha \end{bmatrix} \begin{bmatrix} \cos\beta & -\sin\beta & 0 \\ \sin\beta & \cos\beta & 0 \\ 0 & 0 & 1 \end{bmatrix} \tag{6}$$

Equations 1a, 1b, 1c are expressed in closed form since the aerodynamic angles $(\alpha, \beta)$ and force components $(D, C, L)$ are functions of the aircraft's state variables and external conditions. According to Figure 8a, the state variables $(u, v, w)$, which are components of the aircraft's velocity vector $\mathbf{V}$ in $\mathcal{F}_B$, are related to $(\alpha, \beta)$ as follows:

$$u = V \cos\beta \cos\alpha \tag{7a}$$
$$v = V \sin\beta \tag{7b}$$
$$w = V \cos\beta \sin\alpha \tag{7c}$$

where

$$V = \sqrt{u^2 + v^2 + w^2} \tag{8}$$

The instantaneous angles of attack and sideslip are given by:

$$\alpha = \tan^{-1}\frac{w}{u}, \quad \beta = \sin^{-1}\frac{v}{\sqrt{u^2 + v^2 + w^2}} \tag{9}$$

The aerodynamic forces are described using their aerodynamic coefficients in the following standard formulas:

$$D = \frac{1}{2}\rho V^2 S C_D, \quad C = \frac{1}{2}\rho V^2 S C_C, \quad L = \frac{1}{2}\rho V^2 S C_L \tag{10}$$

where the air density $\rho$ depends on the flight altitude $h = -z_{E,G}$ and other atmospheric properties like the sound speed $a$ [70]. $S$ represents a reference area, while the coefficients $(C_D, C_C, C_L)$ vary with the aircraft's state and external inputs.

Finally, as shown in Figure 8b, the thrust force $F^{(T)}$ of magnitude $T$ is expressed in the body-frame components as follows:

$$\begin{Bmatrix} F_x^{(T)} \\ F_y^{(T)} \\ F_z^{(T)} \end{Bmatrix} = \delta_T T_{\max}(h, M) \begin{Bmatrix} \cos \mu_T \\ 0 \\ \sin \mu_T \end{Bmatrix} \tag{11}$$

where $\mu_T$ is a constant angle between the thrust line and the reference axis $x_B$ in the aircraft's symmetry plane. The thrust $T = \delta_T T_{\max}(h, M)$, where $\delta_T$ is the throttle setting (an external input), and $T_{\max}(h, M)$ is the maximum thrust available, dependent on altitude and Mach number $M = V/a$.

The conservation of angular momentum equations (CAMEs) for a rigid aircraft with constant mass are given by [9]:

$$\dot{p} = (C_1 r + C_2 p)q + C_3 L + C_4 N \tag{12a}$$

$$\dot{q} = C_5 pr - C_6(p^2 - r^2) + C_7 M \tag{12b}$$

$$\dot{r} = (C_8 p - C_2 r)q + C_4 L + C_9 N \tag{12c}$$

where

$$C_1 = \frac{1}{\Gamma}[(I_{yy} - I_{zz})I_{zz} - I_{xz}^2], \tag{13a}$$

$$C_2 = \frac{1}{\Gamma}[(I_{xx} - I_{yy} + I_{zz})I_{xz}], \tag{13b}$$

$$C_3 = \frac{I_{zz}}{\Gamma}, \quad C_4 = \frac{I_{xz}}{\Gamma}, \quad C_5 = \frac{I_{zz} - I_{xx}}{I_{yy}}, \tag{13c}$$

$$C_6 = \frac{I_{xz}}{I_{yy}}, \quad C_7 = \frac{1}{I_{yy}}, \tag{13d}$$

$$C_8 = \frac{1}{\Gamma}[(I_{xx} - I_{yy})I_{xx} + I_{xz}^2], \quad C_9 = \frac{I_{xx}}{\Gamma} \tag{13e}$$

and $\Gamma = I_{xx}I_{zz} - I_{xz}^2$ are constants derived from the aircraft's inertia matrix relative to the axes of $\mathcal{F}_B$.

The systems of equations 1, 12 for CLMEs and CAMEs projected onto the moving frame $\mathcal{F}_B$ must be supplemented with additional equations to fully describe the aircraft dynamics and evolve its state over time. One such set of equations is the flight path equations (FPEs), which describe the aircraft's trajectory relative to the Earth-based inertial frame. These equations yield the instantaneous position $\{x_{E,G}(t), y_{E,G}(t), z_{E,G}(t)\}$ of the aircraft's center of gravity $G$ in $\mathcal{F}_E$. The 2D version $\{x_{E,G}(t), y_{E,G}(t)\}$ of the FPEs defines the ground track relative to the aircraft's flight path.

The flight path equations (FPEs) are derived by transforming the vector $\mathbf{V}$ from frame $\mathcal{F}_B$ to frame $\mathcal{F}_E$:

$$\begin{Bmatrix} \dot{x}_{E,G} \\ \dot{y}_{E,G} \\ \dot{z}_{E,G} \end{Bmatrix} = [T_{EB}] \begin{Bmatrix} u \\ v \\ w \end{Bmatrix} \tag{14}$$

with $[T_{EB}] = [T_{BE}]^{\mathrm{T}}$ as defined in equation 3. The matrix form of the FPEs is:

$$\begin{Bmatrix} \dot{x}_{E,G} \\ \dot{y}_{E,G} \\ \dot{z}_{E,G} \end{Bmatrix} = \begin{bmatrix} q_0^2 + q_x^2 - q_y^2 - q_z^2 & 2(q_x q_y + q_0 q_z) & 2(q_x q_z - q_0 q_y) \\ 2(q_x q_y - q_0 q_z) & q_0^2 - q_x^2 + q_y^2 - q_z^2 & 2(q_y q_z + q_0 q_x) \\ 2(q_x q_z + q_0 q_y) & 2(q_y q_z - q_0 q_x) & q_0^2 - q_x^2 - q_y^2 + q_z^2 \end{bmatrix} \begin{Bmatrix} u \\ v \\ w \end{Bmatrix} \tag{15}$$

The inputs for the FPEs are the aircraft's attitude quaternion components along with the components $(u, v, w)$, which are derived from the combined CLMEs and CAMEs system.

The rigid-body kinematic equations (KEs) using the aircraft's attitude quaternion components [9] are expressed in matrix form as:

$$\begin{Bmatrix} \dot{q}_0 \\ \dot{q}_x \\ \dot{q}_y \\ \dot{q}_z \end{Bmatrix} = \frac{1}{2} \begin{bmatrix} 0 & -p & -q & -r \\ p & 0 & r & -q \\ q & -r & 0 & p \\ r & q & -p & 0 \end{bmatrix} \begin{Bmatrix} q_0 \\ q_x \\ q_y \\ q_z \end{Bmatrix} \tag{16}$$

The inputs to these KEs are the angular velocity components $(p, q, r)$ in $\mathcal{F}_B$, and solving these equations provides the kinematic state variables $(q_0, q_x, q_y, q_z)$.

The system comprising (CLMEs)-(CAMEs)-(FPEs)-(KEs), i.e., 1, 12, 15, and 16, represents a complete set of 13 coupled nonlinear differential equations that describe the 6-DoF rigid-body dynamics of atmospheric flight. These equations are in closed form once the aerodynamic and propulsive external forces and moments are fully modeled as functions of the 13 state variables:

$$\mathbf{x} = [u, v, w, p, q, r, x_{E,G}, y_{E,G}, z_{E,G}, q_0, q_x, q_y, q_z]^{\mathrm{T}} \tag{17}$$

This state vector $\mathbf{x}$, along with various external inputs grouped into an input vector, commonly referred to as $\mathbf{u}$, fully characterizes the system.

The F-16 public domain model utilized in this study includes a sophisticated and high-fidelity flight control system (FCS). The FCS, which incorporates state feedback from the aircraft dynamics block, consists of the following channels: (i) Roll command $\delta_a$ (affecting right aileron deflection angle $\delta_a$ and antisymmetric left aileron deflection), (ii) Pitch command $\delta_e$ (controlling elevon deflection angle $\delta_e$), (iii) Yaw command $\delta_r$ (manipulating rudder deflection angle $\delta_r$), (iv) Throttle lever command $\delta_T$ (adjusting throttle setting $\delta_T$ and enabling jet engine afterburner).

## A.2  Task Scenarios

The task scenarios can be categorized by objectives into *Heading, Control, and Tracking*. (1) *Heading*: The objective is to control the fixed-wing aircraft to reach a predetermined altitude, yaw angle, and speed within a specified time. This task serves as the foundation for multi-aircraft collaboration and pursuit tasks. (2) *Control*: The objective is to control the fixed-wing aircraft to reach a predetermined pitch angle, yaw angle, and speed within a specified time. This task serves as the fundamental control basis for fixed-wing aircraft trajectory tracking. (3) *Tracking*: The objective is to control the fixed-wing aircraft to reach a predetermined coordinate position (in the geocentric coordinate system) within a specified time. This work designs a hierarchical control algorithm for this task. The lower-level controller is capable of completing the Control task, while the upper-level planner algorithm aims to achieve the overall task objective. This task forms the basis for performing aerobatic maneuvers with fixed-wing aircraft.

The task scenarios can also be categorized by flight conditions into HighSpeed, HighAltitude, Windy, and Noisy. (1) *HighSpeed*: Control of high-maneuverability flight of fixed-wing aircraft under high-speed conditions (speed exceeding Mach 1). (2) *HighAltitude*: Control of high-maneuverability flight of fixed-wing aircraft under high-altitude conditions (altitude exceeding 30,000 feet). (3) *Windy*: Control of high-maneuverability flight of fixed-wing aircraft under windy conditions. (4) *Noisy*: Control of high-maneuverability flight of fixed-wing aircraft when there is noise in the observation measurements.

We design different environment rewards for different task objectives. For the Heading and Tracking tasks, the environment reward is the negative Euclidean norm (L2 norm) error between the current state and the target state. For the Control task, the environment reward is the negative optimal rotation angle from the current attitude to the target attitude. We also designed various termination conditions and terminal rewards for different tasks, as shown in Table 5.

Table 5: Termination conditions and terminal rewards for different tasks.

| Name | Key Value | Description | Terminal reward |
|---|---|---|---|
| ExtremeState | AOA, AOS | AOA and AOS exceeding limit ranges. | -200 |
| HighSpeed | TAS | speed exceeding Mach 3. | -200 |
| LowAltitude | altitude | altitude falling below 2500 feet. | -200 |
| LowSpeed | TAS | speed falling below Mach 0.01. | -200 |
| overload | G | G exceeding 10. | -200 |
| UnreachTarget | $\mathbf{x_t} - \mathbf{x_{target}}$ | the target is not reached. | -200 |
| ResetTarget | $\mathbf{x_t} - \mathbf{x_{target}}$ | the target is successfully reached. | 200 |

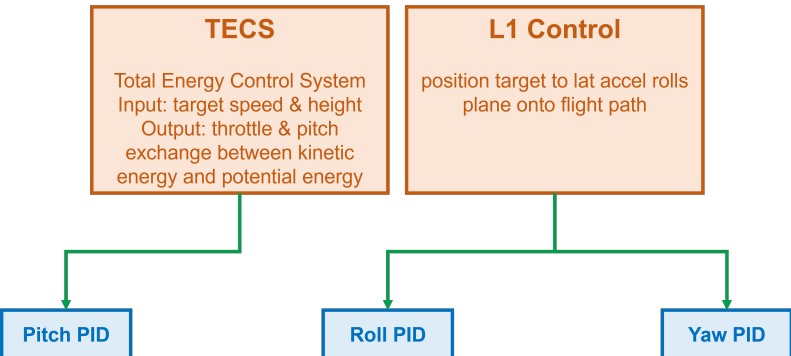

Figure 9: The control system structure for traditional methods.

### A.3 Baseline Libraries

**Traditional Methods** These are based on open-source fixed-wing aircraft control algorithms from the Ardupilot platform, using a hierarchical control approach. The upper layer includes the TECS controller [68], which manages the aircraft's total flight energy by adjusting throttle and pitch to maintain desired altitude and speed, and the L1 controller [69], which manages the flight path by adjusting roll and yaw to follow waypoints or desired path characteristics. The lower layer consists of an attitude loop controller using a dual-loop PID algorithm to control the aircraft's surfaces and achieve three-axis attitude tracking. The control system structure for traditional methods is shown in Figure 9.

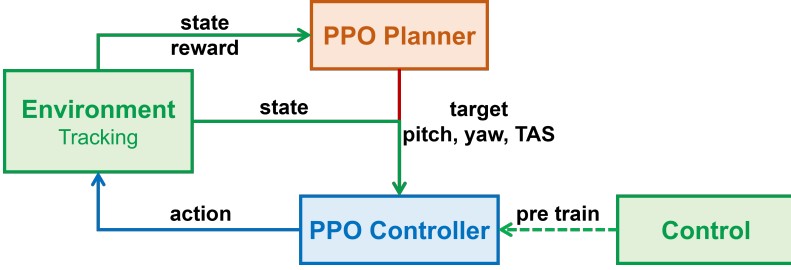

Figure 10: The structure for hierarchical RL method.

**RL Methods** We use PPO for Heading and Control tasks in fixed-wing aircraft. For the Tracking task, we use a hierarchical RL method: the upper-level algorithm converts the target location into desired pitch, yaw, and speed, while the lower level uses the trained PPO algorithm to control the aircraft's surfaces. The structure for hierarchical RL method is shown in Figure 10.

The PPO algorithm's parameter settings are as follows: the learning rate is set to $3 \times 10^{-4}$, the number of PPO epochs is 16, the clipping parameter is 0.2, the maximum gradient norm is 2, and the entropy coefficient is $1 \times 10^{-3}$. Additionally, the hidden layer sizes for the neural networks are set to "128 128", and the recurrent hidden layer size is 128 with a single recurrent layer.

## A.4 Evaluation Metrics

We provide two types of performance evaluation metrics to assess the algorithm's performance of fixed-wing aircraft control: maneuverability indicators and safety indicators. The complete set of evaluation metrics is shown in Table 6.

Table 6: Performance metrics to assess the algorithm's performance of fixed-wing aircraft control.

| Type | Name | Description |
|---|---|---|
| | G | Average G-force during flight. |
| | TAS | Average True Air Speed during flight. |
| | RoC | Average Rate of Climb during flight. |
| | AOA | Average Angle of Attack during flight. |
| Maneuverability Indicators | AOS | Average Angle of Sideslip during flight. |
| | t | Average time to complete the task objective. |
| | P | Average roll rate around the body-fixed x-axis. |
| | Q | Average pitch rate around the body-fixed y-axis. |
| | R | Average yaw rate around the body-fixed z-axis. |
| | ASM | Altitude Safety Margin. |
| | SSM | Speed Safety Margin. |
| | OSM | Overload Safety Margin. |
| Safety Indicators | AOASM | Angle of Attack Safety Margin. |
| | AOSSM | Angle of Sideslip Safety Margin. |
| | FSM | Smoothness of the aircraft's flight state. |

## A.5 Code Structure

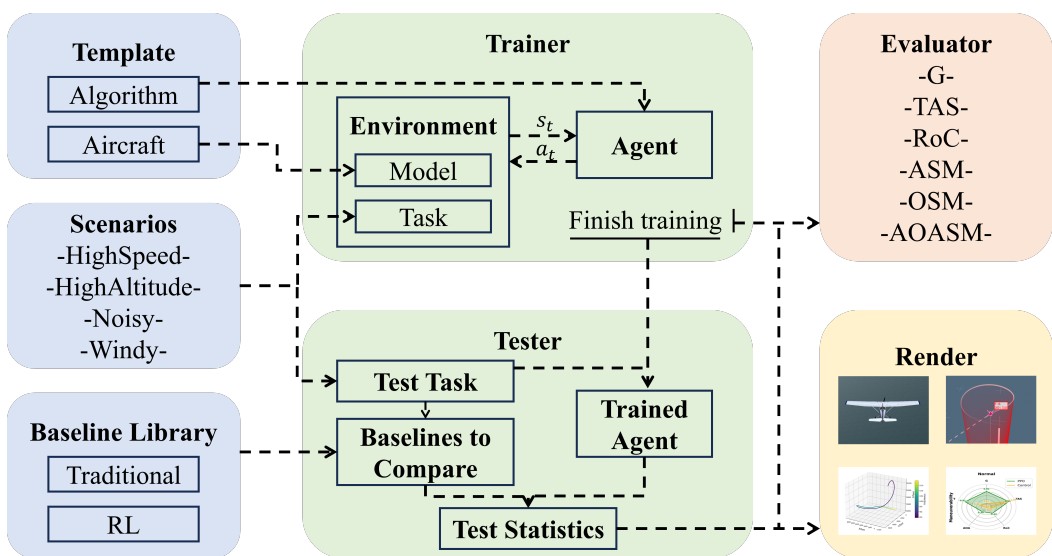

Figure 11: The overall code framework and workflow of NeuralPlane.

The overall code framework and workflow of the platform are illustrated in Figure 11. We also provide a complete algorithmic process for training, testing, and evaluating RL algorithms on the platform. Once the appropriate parameters are selected, the platform can automatically execute the algorithm training, testing, and evaluation processes.

The pseudo-code for the Trainer is detailed in Algorithm 1. With a user-designed learnable agent $A$, a user-specified FDM $M$, and a user-specified task scenario $T$, the NeuralPlane first initializes the environment by determining task scenario and FDM. MetaBox then iteratively trains on each instance

---

**Algorithm 1** NeuralPlaneTrainer

---

**Require:** User-designed learnable agent $A$, user-specified FDM $M$, user-specified task scenario $T$
**Ensure:** Trained Agent $A$, training records
 1: Initialize environment with FDM and task scenario $Env = $ Env_Initialize$(M, T)$;
 2: **while** max learning steps Not reached **do**
 3:     $A$.train_episode$(Env)$;
 4:     Record training data;
 5:     Plot training figures;
 6: **end while**
 7: Summarize and visualize the training records in *Logger* and return the trained agent;

---

**Algorithm 2** TrainingEpisode

---

**Require:** User-designed learnable agent $A$, Constructed environment $Env$
**Ensure:** Training records
 1: $state = Env.\text{reset}()$;
 2: **while** termination condition Not achieved **do**
 3:     $action = A.\text{get\_action}(state)$;
 4:     $next\_state, reward, done, info = Env.\text{step}(action)$;
 5:     Store transition $\langle state, action, reward, done, next\_state \rangle$;
 6:     Update agent $A$;
 7:     Record training data and plot figures;
 8:     $state = next\_state$;
 9: **end while**
10: Summarize training records and return;

---

until the maximum learning steps are reached. For each instance, agent $A$ calls the `train_episode()` function to interact with $Env$ and perform the training. All training logs are managed by the *Logger*.

Next, we focus on the `train_episode()` function. In Algorithm 2, we present a straightforward example of implementing RL training algorithms within `train_episode()`. Starting from $Env$ initialization, in each step, agent $A$ provides $Env$ with actions based on the current state, receives the next state, reward, and other information, and updates the policy accordingly. Within the `env.step()` interface, actions are translated into configurations applied to the aircraft. Rewards and subsequent states are calculated, with logging information summarized concurrently.

For the Tester and Evaluator shown in Algorithm 3, the environment is first initialized to evaluate each algorithm in the set (including several baseline agents and the user's trained agent). The `rollout_episode()` interface is similar to `train_episode()`, but it does not include the policy update procedures. Finally, NeuralPlane evaluates the algorithm's performance and provides visualization of fixed-wing aircraft flight trajectories based on the flight data generated from testing.

---

**Algorithm 3** NeuralPlaneTester and NeuralPlaneEvaluator

---

**Require:** User-specified algorithm set $B$ including baselines and user's trained agent, user-specified
      FDM $M$, User-specified task scenario $T$
**Ensure:** Testing results
 1: Initialize environment with FDM and task scenario $Env = $ Env_Initialize$(M, T)$;
 2: **for** each algorithm $alg \in B$ **do**
 3:     $alg.\text{rollout\_episode}(Env)$;
 4:     Record testing data;
 5:     Plot testing figures;
 6: **end for**
 7: Summarize testing results and call *Evaluator* for standardized metrics and visualization;

---

# B Details of Experiments

## B.1 Experimental Parameters

Before researchers can use NeuralPlane to complete the full workflow of algorithm training, testing, evaluation, and replay, two preliminary steps must be completed: 1) Determine the fixed-wing aircraft dynamics model to be used, the task objectives that the algorithm will control the aircraft to achieve, and the operational conditions. This step is used to initialize the basic parameters of NeuralPlane's core simulation environment. 2) Determine the maximum number of training steps (M), the number of parallel rollouts during training (n), and the number of steps per rollout in one iteration (m). After setting these parameters, the platform can automatically execute the complete process. The experimental parameter settings for different task scenarios are shown in Table 7.

Table 7: The experimental parameter settings for different task scenarios

| Name | n | m | M | env | scenario | model |
|------|------|------|---------------------|----------|----------|-------|
| Heading | 3000 | 3000 | $1.35 \times 10^9$ | Control | heading | F16 |
| Control | 3000 | 3000 | $2.25 \times 10^9$ | Control | control | F16 |
| Tracking | 10000 | 100 | $3 \times 10^8$ | Planning | tracking | F16 |

## B.2 Additional Experimental Results

We conducte multiple experiments across all task scenarios, thoroughly demonstrating NeuralPlane's superiority in supporting RL algorithm training and showcasing the powerful capabilities of RL algorithms in fixed-wing aircraft control. Some experimental results are shown in Figure 12, 13, 14, with all results available at `https://github.com/xuecy22/NeuralPlane`. The results also indicate that in some high-difficulty scenarios, the control effectiveness of RL algorithms needs improvement, highlighting the platform's value and potential for RL research.

We also test the RL algorithms across all task scenarios and perform a visual evaluation of their performance. Some visualization results are shown in Figure 15, with all experimental results available at `https://github.com/xuecy22/NeuralPlane`.

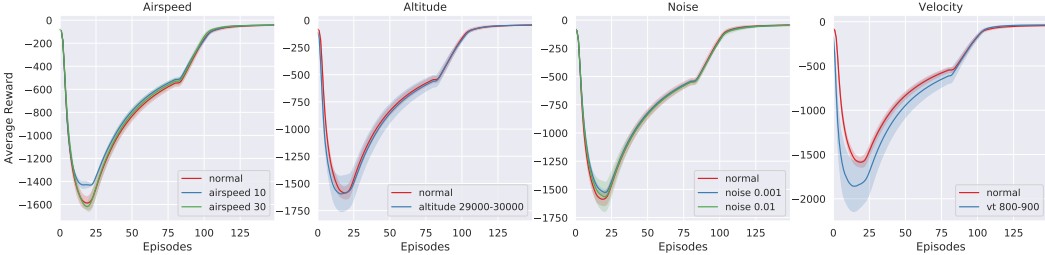

Figure 12: Training curves of PPO in different task scenarios. **From left to right**, the task conditions are different wind speeds, different flight altitudes, different environmental noise levels, and different flight speeds, with the task objective being the Heading task in all cases.

# C Used Assets

Table 8 lists the resources and assets used in NeuralPlane along with their respective licenses. We strictly adhere to these licenses during the development of NeuralPlane.

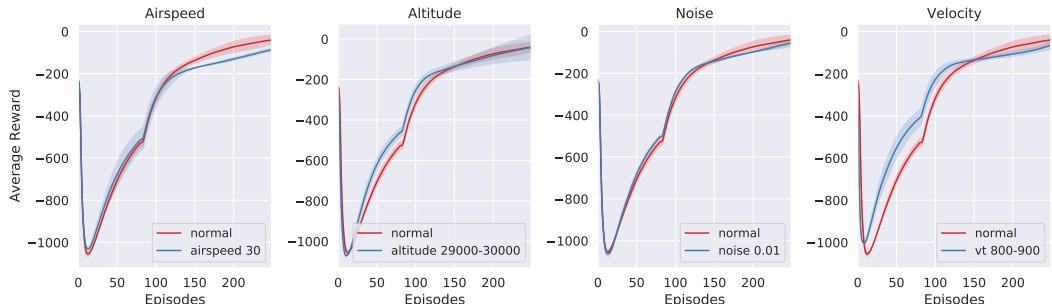

Figure 13: Training curves of PPO in different task scenarios. **From left to right**, the task conditions are different wind speeds, different flight altitudes, different environmental noise levels, and different flight speeds, with the task objective being the Control task in all cases.

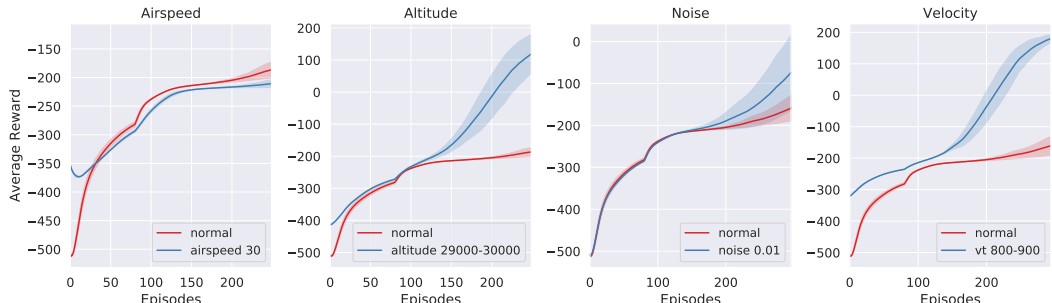

Figure 14: Training curves of PPO in different task scenarios. **From left to right**, the task conditions are different wind speeds, different flight altitudes, different environmental noise levels, and different flight speeds, with the task objective being the Tracking task in all cases.

Table 8: Used assets and their licenses

| Type | Asset | Codebase | License |
| --- | --- | --- | --- |
| Baseline | PPO [27] | CloseAirCombat [71] | LGPL-3.0 license |
| Platform to Compare | Ardupilot [22] JSBSim [21] QPlane [23] | Ardupilot [22] JSBSim [21] QPlane [23] | LGPL-3.0 license LGPL-2.1 license - |

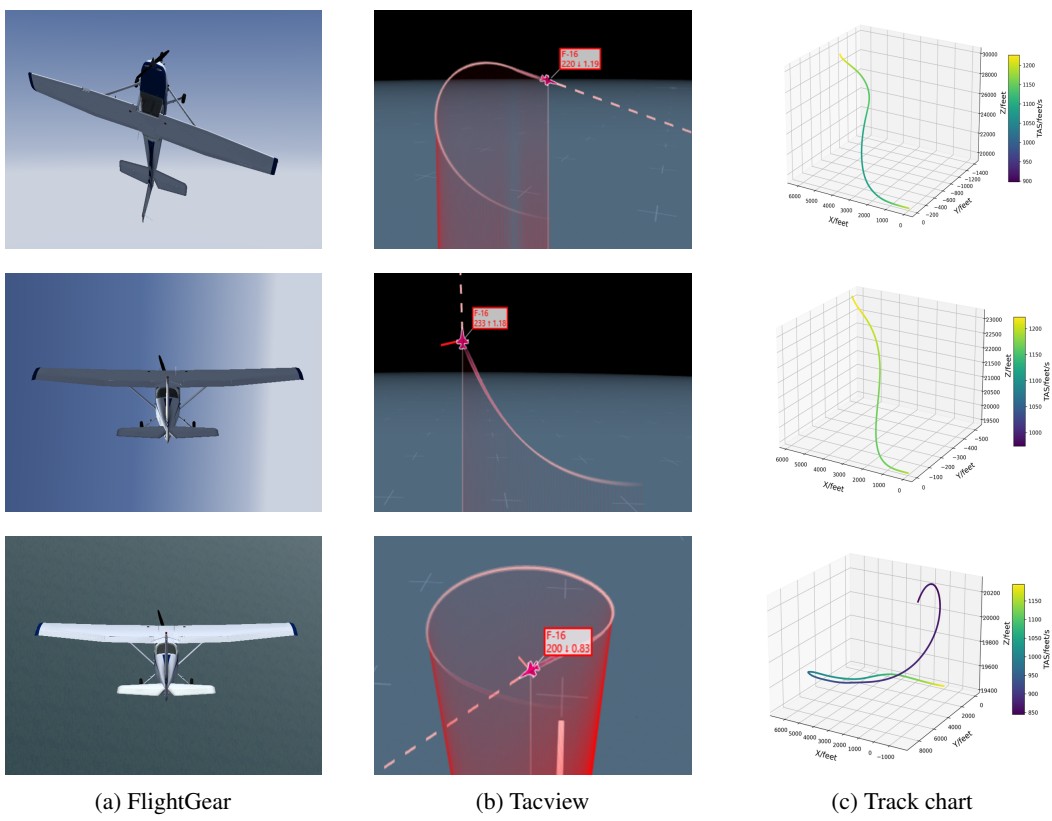

(a) FlightGear       (b) Tacview       (c) Track chart

Figure 15: Visualization of fixed-wing aircraft flight trajectories. **Top**: The results of the Heading task. **Middle**: The results of the Control task. **Bottom**: The results of the Tracking task.

