# OpenReview forum: "NeuralPlane: An Efficiently Parallelizable Platform for Fixed-wing Aircraft Control with Reinforcement Learning"
_NeurIPS.cc/2024/Datasets_and_Benchmarks_Track — NeurIPS 2024 Track Datasets and Benchmarks Poster_

### Official Review · Reviewer_HNDj · 2024-07-17
**Flight dynamics model for aircraft control**

**Rating:** 6
**Confidence:** 3
**Correctness:** Not all details have been checked but…
**Clarity:** The paper is clearly written.

**Review:**

- The simulator achieves significant speedups compared to previous simulators that are currently available.
- It provides a range of task scenarios and is compatible with various reinforcement learning algorithms.
- The provided experiments demonstrate that the simulator can be easily parallelized even though the GPU memory requirements increase quickly.

**Strengths:**

- This is an interesting contribution that can serve as a simulator platform for a wide range of research on control and reinforcement learning of aircraft control.
- The simulator is designed with runtime costs in mind, which is different from many current aircraft simulators that offer many task scenarios but can be slow.

**Additional Feedback:**

No additional feedback besides the above.

**Documentation:**

User friendliness of the simulator could be improved. This includes clearer documentation but also improved interfaces.

**Ethics:**

No concerns.

**Limitations:**

The authors mention several limitations. In particular the low user friendliness is a weakness of the simulator.

**Opportunities For Improvement:**

- The user friendliness is low, which I consider a major weakness. (The authors mention this as a limitation.) User friendliness is key for adoption of the simulator.
- Provide more task scenarios.

**Relation To Prior Work:**

Previous work is discussed and the differences are made clear. Especially the GPU support is different from previous simulators.

**Summary And Contributions:**

The authors introduce a flight dynamics model for aircraft control with reinforcement learning.

---

> ### Author Response · Authors · 2024-08-17
> **Authors' response to Reviewer HNDj**
>
> Thank you for the valuable feedback and questions. We will address your concerns in the following parts:
>
> **Q1: The user friendliness is low.**
>
> We have developed comprehensive tutorials to guide users in using NeuralPlane and configuring its parameters. These tutorials are beginner-friendly and require no prior RL knowledge. We are committed to updating NeuralPlane and its tutorials to support the RL community. Additionally, we plan to refine NeuralPlane’s code interface and framework to enhance usability further.
>
> **Q2: Provide more task scenarios.**
>
> We have introduced several new tasks for fixed-wing aircraft, including continuous multi-target tracking, aerobatic maneuvers like ”figure-8” flight, and multi-aircraft collaboration. These additions might highlight the strong potential of RL in this field. The extra tasks are now available in NeuralPlane's [code repository](https://anonymous.4open.science/r/NeuralPlane/README.md).
>
> Additionally, we are integrating a gust model into the simulation environment to replicate more complex atmospheric conditions. This addition aims to diversify the range of missions and expand the scope of RL applications. We are also exploring the creation of a continuous multi-target tracking racing scenario and plan to further our efforts by organizing RL racing competitions for fixed-wing aircraft in the future.

---

> ### Author Response · Authors · 2024-08-30
> **Authors' response to Reviewer HNDj**
>
> Dear reviewer,
>
> Thank you again for taking the time to review our work. Please let us know if you have any concerns left after our response. Looking forward to your valuable feedback.

---

### Official Review · Reviewer_nViW · 2024-07-25
**Review of the paper**

**Rating:** 8
**Confidence:** 3
**Correctness:** The claims made in the submission is …
**Clarity:** Yes, the paper is well written.

**Review:**

NeuralPlane represents a significant advancement in the field of fixed-wing aircraft control by providing a scalable, high-fidelity simulation platform tailored for RL applications. Its efficient parallelization capabilities, comprehensive benchmarking features, and practical implementation support make it a valuable tool for researchers and developers aiming to push the boundaries of autonomous flight control systems. Despite some implementation complexities and generalization challenges, NeuralPlane stands poised to accelerate the development and deployment of RL-based solutions in aerospace applications.

**Strengths:**

1, NeuralPlane leverages GPU-accelerated FDM computations to achieve a remarkable single-step simulation time of 0.2 seconds for up to 1 million parallel aircraft instances. This scalability surpasses existing fixed-wing aircraft simulation platforms, facilitating large-scale RL training and evaluation.

2, It provides a diverse set of fixed-wing aircraft task scenarios and baseline algorithms, enabling researchers to evaluate RL methods across various problem contexts. This includes interfaces for both RL and traditional control methods, fostering detailed performance analysis and comparison.

3, The platform includes clear code templates, frameworks, and visualization tools for easy integration, training, testing, and evaluation of algorithms. This accessibility lowers the barrier for researchers and practitioners to utilize and extend the platform for their specific needs.

**Additional Feedback:**

None

**Documentation:**

Yes, it is sufficient detail on Documentation.

**Limitations:**

The same as the Opportunities For Improvement.

**Opportunities For Improvement:**

Overall, the paper is good. These are some minor problem.

1, While NeuralPlane offers comprehensive support and tools, the initial setup and configuration of GPU-accelerated simulations might require specialized knowledge in both RL algorithms and GPU programming.

2, Although NeuralPlane enhances scalability and fidelity, challenges in generalizing RL-trained models across diverse environmental conditions and real-world scenarios may persist.

**Relation To Prior Work:**

The paper clearly discussed the related work.

**Summary And Contributions:**

The paper introduces NeuralPlane, a novel platform designed for efficient and scalable parallel simulations of fixed-wing aircraft control using reinforcement learning (RL). Traditional flight control methods for fixed-wing aircraft often struggle with complex nonlinear dynamics and environmental disturbances, necessitating advanced techniques like RL for agile and stable flight. NeuralPlane addresses these challenges by offering high-fidelity simulations via GPU-accelerated Flight Dynamics Models (FDM), achieving unprecedented simulation efficiency at a scale of up to 1 million aircraft instances.

---

> ### Author Response · Authors · 2024-08-17
> **Authors' response to Reviewer nViW**
>
> Thank you for the valuable feedback and questions. We will address your problems in the following parts:
>
> **Q1: Tutorials for the initial setup and configuration of GPU-accelerated simulations.**
>
> We have created detailed tutorials on using NeuralPlane and configuring its parameters. These tutorials are beginner-friendly and do not require specific RL knowledge, making them accessible to a wider audience. We are committed to continuously updating and maintaining NeuralPlane and its tutorials to support the RL community.
>
> **Q2: Further challenges for RL-trained models in multi-task generalization or sim2real.**
>
> We acknowledge that while NeuralPlane is a highly parallelizable and high-fidelity RL training platform, challenges with sim-to-real transfer still remain. Our primary goal with NeuralPlane is to create a robust simulation environment that advances the application of RL in fixed-wing aircraft, ultimately leading to successful real-world deployment.
>
> To address sim-to-real challenges, we incorporated various tasks in NeuralPlane to support researchers. For instance, we developed a gust model to simulate more realistic atmospheric conditions and introduced noise into environmental observations to mimic potential disturbances in real fixed-wing aircraft systems. Additionally, to tackle multi-task generalization, we designed migration experiments across different tasks to evaluate and confirm the algorithm’s generalization capabilities, as shown in Figure 7.

---

> ### Author Response · Authors · 2024-08-30
> **Authors' response to Reviewer nViW**
>
> Dear reviewer,
>
> Thank you again for taking the time to review our work. Please let us know if you have any concerns left after our response. Looking forward to your valuable feedback.

---

### Official Review · Reviewer_YwUv · 2024-07-31
**GPU accelerated flight RL environment**

**Rating:** 6
**Confidence:** 3
**Correctness:** The work seems correct

**Review:**

The introduced environment will help RL researchers in the FDM domain to run more extensive experiments than with other FDM platforms on the same compute budget.
There is no question that some RL research will find this work very useful.

The main limitations I see are
- the lack of flight tasks/objectives: The works seem to be very general in terms of tasks that can be defined by the paper itself and lack a more detailed discussion on what *novel* flight tasks/objectives can be studied with this simulator. Without this, I feel the paper has less value for RL (i.e. pushing the boundary of what's possible with RL)
- the narrow focus of the work. Particularly, I think this work might be more interesting for robotics and aero/astro research groups than the ML/RL audience of NeurIPS due to the very selective type of simulation

**Strengths:**

- Fast environment, making RL research much more accessible
- The environment supports headless mode and connections to different visualization platforms.

**Additional Feedback:**

no

**Clarity:**

The work is mostly well written, though a grammar check would be a good idea here and there.

**Documentation:**

yes

**Limitations:**

There are some obvious military use cases of the introduced FDM that they authors should potentially mention.

**Opportunities For Improvement:**

- Add novel flight tasks/RL objectives that are not possible with existing FDM environments

**Relation To Prior Work:**

The paper could benefit from a discussion with a general theme currently happening in RL research of creating GPU-accelerated environments to speedup RL experiment times.

**Summary And Contributions:**

The paper introduces a GPU-accelerated flight RL environment. The environment allows running millions of simulations in parallel, thus enabling the collection of a large number of rollouts for RL training within a short time.

---

> ### Author Response · Authors · 2024-08-17
> **Authors' response to Reviewer YwUv**
>
> Thank you for the valuable feedback and suggestions. We will address your problems in the following parts:
>
> **Q1: The lack of flight tasks/objectives.**
>
> We have introduced several new tasks for fixed-wing aircraft, including continuous multi-target tracking, aerobatic maneuvers (e.g., “figure-8” flight), and multi-aircraft collaboration. These tasks highlight the strong performance of RL in this domain. We've updated NeuralPlane's [code repository](https://anonymous.4open.science/r/NeuralPlane/README.md) to include these additions, reflecting our commitment to continually enhancing the platform.
>
> We are also integrating a gust model into the simulation environment to simulate more complex atmospheric conditions. We believe this will further enrich the mission variety and expand the possibilities for RL applications. Additionally, we are considering developing a continuous multi-target tracking racing scenario and plan to extend our work by hosting RL racing competitions for fixed-wing aircraft in the future.
>
> **Q2: The narrow focus of the work.**
>
> Thank you for recognizing our work as highly interesting for robotics and aero/astro research groups. We believe that NeuralPlane will not only appeal to these groups but also be of great interest to researchers in the RL community.
>
> On the one hand, providing a fixed-wing aircraft simulation platform that supports massively parallel RL training is a significant contribution to the RL community. This platform enables researchers to expand the application scenarios of their algorithms, and we are confident that RL will achieve even greater advancements in the field of fixed-wing aircraft.
>
> On the other hand, the inherent characteristics of fixed-wing aircraft—such as strong nonlinearity, high degrees of freedom, environmental randomness, and strict safety requirements—make them particularly well-suited for RL research. We designed a continuous target tracking task to facilitate the study and validation of goal-conditioned RL algorithms. The gusty wind environment and noise-affected tasks we introduced are ideal for researching and validating robust RL algorithms. Additionally, our multi-aircraft cooperative task is designed to validate multi-agent RL algorithms, and the safety requirements of fixed-wing aircraft naturally align with the focus of safety RL research.
>
> | task | RL algorithm |
> | --- | --- |
> | continuous target tracking task | goal-conditioned RL |
> | gusty wind environment | robust RL |
> | noise-affected task | robust RL |
> | multi-aircraft cooperative task | multi-agent RL |
> | safety requirements | safety RL |
>
> Therefore, we believe that NeuralPlane will not only advance RL-based methods in the fixed-wing aircraft domain but also significantly contribute to algorithm design and application in the broader RL community.
>
> **Q3: Discussion about creating GPU-accelerated environments to speedup RL experiment times.**
>
> Running physics simulations on GPUs can lead to significant speedups, which is crucial for RL training. In recent years, hardware-accelerated simulation platforms have greatly advanced the RL field. For example, NVIDIA's Isaac Gym[1] enables high-performance training for various robotics tasks directly on GPUs. In autonomous driving, platforms like GPUDrive and Waymax offer similar benefits. GPUDrive[2], built on the Madrona Game Engine, can generate over a million steps per second, allowing for fast and effective RL training using the Waymo Motion dataset. Waymax[3], designed for large-scale simulation in multi-agent scenarios, further advances autonomous driving research.
>
> Another example is Pgx[4], a suite of board game RL environments optimized for GPU/TPU accelerators, which simulates environments 10-100 times faster than existing Python implementations. These developments underscore the importance of building RL training platforms that support massively parallel simulations, as they can significantly enhance the speed and effectiveness of RL algorithms across various domains. We believe that NeuralPlane can also advance RL in the fixed-wing aircraft domain.
>
> [1] Makoviychuk V, Wawrzyniak L, Guo Y, et al. Isaac gym: High performance gpu-based physics simulation for robot learning. arXiv preprint arXiv:2108.10470, 2021.
>
> [2] Kazemkhani S, Pandya A, Cornelisse D, et al. GPUDrive: Data-driven, multi-agent driving simulation at 1 million FPS. arXiv preprint arXiv:2408.01584, 2024.
>
> [3] Gulino C, Fu J, Luo W, et al. Waymax: An accelerated, data-driven simulator for large-scale autonomous driving research. Advances in Neural Information Processing Systems, 2024, 36.
>
> [4] Koyamada S, Okano S, Nishimori S, et al. Pgx: Hardware-accelerated parallel game simulators for reinforcement learning. Advances in Neural Information Processing Systems, 2024, 36.

---

> ### Author Response · Authors · 2024-08-30
> **Authors' response to Reviewer YwUv**
>
> Dear reviewer,
>
> Thank you again for taking the time to review our work. Please let us know if you have any concerns left after our response. Looking forward to your valuable feedback.

---

### Official Review · Reviewer_EGTs · 2024-07-31
**NeuralPlane: a framework to simulate, train and assess fixed-wing aircraft control**

**Rating:** 7
**Confidence:** 3
**Correctness:** The paper seems to be correct.
**Clarity:** The paper is clearly written.

**Review:**

I believe the paper describes a valuable tool. As a community, the RL researchers are eager to get more parallelizable simulation engines such as this one and it's great to see that domains such as aircraft control can benefit from that.

The paper is well written: I'm far from being an expert in the field, yet I could understand it all to a decent degree without having to look for information outside the manuscript.
The need for this tool is also clearly stated, which is nice for readers who are not familiar with this problem.

**Strengths:**

I liked that the training curves had a low error bar, and I wonder what this is due to? Is the initialization always the same? Usually, even if they converge to a similar results, PPO runs tend to vary wildly.

I also liked the idea of using an MLP instead of the lookup table and I've found the tests convincing for this substitution. Is that a separate product? Perhaps people could use that without using NeuralPlane, or that would make no sense? Are the weights of the MLP provided with the library? Finally, what are the specificities of the MLP?

**Additional Feedback:**

NA

**Documentation:**

The experiments are clearly documented.

**Limitations:**

The opportunities for improvement are clearly stated.

**Opportunities For Improvement:**

The reason why the three steps (simulation, training, testing) have been integrated in a single tool is not clearly stated. I imagine it is to facilitate adoption and reduce friction between frameworks, but it could also narrow the scope on the long term. I think the paper would benefit from a bit of discussion on the topic.

For instance, the question in italic on p2 comes a bit out of the blue, and the message there could be massaged a bit more.

The choice of RL algos is arbitrary but I think that's ok. I'm more worried about how difficult it would be to implement a new one. What tools are the framework using? Is it python, c++, else? Does it rely on a third party implementation, eg SB3, RLLib, TorchRL? A bit more technical details in the manuscript would help!

For a work like this, a sentence about future maintenance (or adoption?) would help too!

The paper does not talk about deployment on real aircraft, is it something the authors have already considered? Is there a path forward to bridge that gap?

I did not see a reference for flightgear, maybe it does not have one. It would be nice to give some context about what it is (same for TacView).

L122: MDP has not been defined, and has no ref. Also, It isn't really used afterwards so I would consider shortening this paragraph and moving it somewhere else (doesn't help understand the paper). Giving a clear overview of the lib would help more at the beginning of section 3!
L126: "where normailze denote " -> "where normalize denotes "

**Relation To Prior Work:**

The relation to prior work is well established.

**Summary And Contributions:**

The paper presents a new complehensive and efficient framework for fixed-wing aircraft control simulation, training and model assessment.

The focus is on parallelization of the operations using GPUs, RL for training (as well as classical control) and testing (as well as rendering).
The authors make the point that no other framework allows headless training and that other frameworks are not as scalable as NeuralPlane which leverages the highly parallelizable compute power of GPUs.

---

> ### Author Response · Authors · 2024-08-17
> **Authors' response to Reviewer EGTs (1/2)**
>
> Thank you for the valuable feedback and suggestions. We are grateful for your detailed comments about the clarity, references and grammatical mistakes, and we are committed to enhancing the quality and completeness of our work. We will address your questions in the following parts:
>
> **Q1: Why do the training curves have a low error bar?**
>
> In our experiments, each training curve was generated using at least 10 different random seeds to ensure reliability. We set the initial state randomness manually and presented the training results using the default range (details will be provided in the appendix). This guarantees distinct and random initializations.
>
> The low error bars in our training curves are due to the use of 3,000 parallel rollouts during training, which increases the amount of sampled data—a strength of NeuralPlane. We also compared training curves with fewer parallel rollouts and found that using fewer rollouts leads to larger error bars, as shown in Figure 3 (left).
>
> | number of parallel rollouts | training performance |
> | --- | --- |
> | 32 | -81.90$\pm$41.99 |
> | 256 | -43.83$\pm$4.11 |
> | 3000 | -24.01$\pm$1.24  |
>
> Furthermore, for the training of the aircraft control strategy, we have high requirements for the stability of the algorithm. To ensure that the PPO algorithm can achieve stable control, we designed several expert reward functions (we have published them along with the code), which help to improve the training stability.
>
> **Q2: Details about the MLP used to approximate the lookup table.**
>
> We implemented the MLP as a tool to replace traditional table lookups for aerodynamic parameters and provided clear training code and tutorials for researchers. NeuralPlane supports a wide range of fixed-wing aircraft dynamics models and allows for customization. The weights of the MLP are integrated into NeuralPlane's [dynamics model library](https://anonymous.4open.science/r/NeuralPlane/F16_AeroData/model/test_model.py).
>
> The MLP takes specific state variables, such as rudder position and airflow angle, as inputs and outputs the corresponding aerodynamic parameters, similar to the traditional approach. Given the simplicity of fitting these parameters, our MLP uses two hidden layers of size 20 and 10. For more complex data, researchers can expand the network size. We achieved excellent results with just 50 training iterations. More details about the MLP design and training are in the appendix.
>
> By using the MLP instead of table lookups, we eliminate interpolation and round-robin computations, significantly improving computational efficiency in massively parallel computations.
>
> **Q3: Why the three steps (simulation, training, testing) have been integrated in a single tool? How difficult it would be to implement a new RL algorithm?**
>
> We developed NeuralPlane as a Python-based implementation, a generic RL simulation environment built on the standard Gym interface. This environment can be used independently, making it compatible with any RL training framework (e.g., SB3, RLLib). Our training framework uses [MAPPO-like](https://github.com/marlbenchmark/on-policy/) code, designed for easy modification and debugging by researchers. We also included a testing module to visualize simulation results, aiding in training and debugging.
>
> These three modules are integrated to streamline the training process, allowing researchers to complete algorithm training without third-party libraries or frameworks. Additionally, we provided the code for all three modules as a clear template, enabling researchers to further develop them according to their specific needs.

---

> ### Author Response · Authors · 2024-08-17
> **Authors' response to Reviewer EGTs (2/2)**
>
> **Q4: The question in italic on p2 comes a bit out of the blue.**
>
> Our work addresses a critical gap in applying RL to fixed-wing aircraft. Previous simulation platforms lack support for massively parallel simulation, complicating data collection for RL algorithms. Additionally, most existing platforms do not provide high-fidelity simulations, hindering the sim-to-real transfer of RL algorithms.
>
> To overcome these challenges, we developed NeuralPlane, enabling large-scale, high-fidelity parallel simulation for fixed-wing aircraft. We believe this contribution will significantly advance RL applications in the fixed-wing aircraft domain.
>
> **Q5: Future maintenance.**
>
> We promise to continue maintaining this environment in the future, including adding more scenarios, supporting more aircraft models, improving parallel efficiency, and maintaining the GitHub repository, among other improvements.
>
> **Q6: Deployment on real aircraft.**
>
> Our framework supports custom aircraft models. For real aircraft, if we can get the corresponding aerodynamic parameters and follow our new model import process, we can create a parallel simulator for RL training. The strategies learned in this environment can to some extent be transferred to the real world. However, sim-to-real still has many other issues to overcome, such as how to get clear sensor data in real world, which requires additional effort for research.
>
> **Q7: Reference for FlightGear.**
>
> [FlightGear](https://www.flightgear.org) is an open-source software that provides a realistic environment for flight simulation, which also supports rendering trajectories for aircraft controlling. [TacView](https://www.tacview.net) is a commercial software for flight data analysis and rendering. Developing such a visual rendering software for aircraft is too costly, so we choose to use existing softwares to visualize flight paths.

---

> ### Author Response · Authors · 2024-08-30
> **Authors' response to Reviewer EGTs**
>
> Dear reviewer,
>
> Thank you again for taking the time to review our work. Please let us know if you have any concerns left after our response. Looking forward to your valuable feedback.

---

### Author Response · Authors · 2024-09-01
**Summary of Responses**

We sincerely appreciate the reviewers’ thorough evaluation and insightful feedback on our submission. **We are pleased that the reviewers recognized the importance of our contribution, particularly the introduction of NeuralPlane as a scalable, GPU-accelerated environment for reinforcement learning (RL) in fixed-wing aircraft control**.

In response to the reviewers’ suggestions, we have made the following key revisions:

1. **Enhanced Task Scenarios**: We added new flight tasks, such as continuous multi-target tracking, aerobatic maneuvers, and multi-aircraft collaboration. These enhancements demonstrate the broader applicability of RL in aviation and address concerns regarding the initial tasks’ specificity.
2. **Improved Documentation and Accessibility**: To improve NeuralPlane’s usability, we created detailed, beginner-friendly tutorials that do not require prior RL knowledge. We also committed to refining the interface to make it more accessible to a wider audience.
3. **Clarified Integration and Technical Details**: We clarified the integration of simulation, training, and testing into a single tool, emphasizing the streamlined adoption process. Additionally, we expanded on the technical aspects of NeuralPlane’s compatibility with various RL frameworks.
4. **Discussion on GPU-Accelerated Environments**: We included a broader discussion on the role of GPU-accelerated environments in RL research, situating NeuralPlane within this context and referencing relevant studies.
5. **Sim-to-Real Considerations**: We addressed the potential for deploying NeuralPlane-trained models on real aircraft, acknowledging the challenges of sim-to-real transfer and explaining how our platform can support researchers in this area.

These revisions enhance the clarity, usability, and impact of our work. **We remain committed to the continuous improvement of NeuralPlane and look forward to contributing further to the RL and aerospace communities**.

Thank you again for your time and constructive feedback.

---

### Decision · Program_Chairs · 2024-09-26

**Decision:**

Accept (Poster)

**Comment:**

The reviewers and I are in agreement that this new benchmark is ready for acceptance. It provides a new setting and simulation engine for the RL community to train and evaluate methods on in an efficient way, which will also be helpful for the control and aviation communities. During the discussion, there were a few concerns on the lack of diverse flight tasks and accessibility, but these have mostly been addressed during the discussion. We encourage the authors to incorporate all of these suggestions into the final version of the paper and as they continue maintaining the benchmark for the community.